# A High Separation Factor for ^165^Er from Ho for Targeted Radionuclide Therapy

**DOI:** 10.3390/molecules26247513

**Published:** 2021-12-11

**Authors:** Isidro Da Silva, Taylor R. Johnson, Jason C. Mixdorf, Eduardo Aluicio-Sarduy, Todd E. Barnhart, R. Jerome Nickles, Jonathan W. Engle, Paul A. Ellison

**Affiliations:** 1Department of Medical Physics, University of Wisconsin School of Medicine and Public Health, 1111 Highland Avenue, Madison, WI 53705, USA; isidro.dasilva@cnrs-orleans.fr (I.D.S.); trjohnson32@wisc.edu (T.R.J.); jmixdorf@wisc.edu (J.C.M.); aluiciosardu@wisc.edu (E.A.-S.); tebarnhart@wisc.edu (T.E.B.); rnickles@wisc.edu (R.J.N.); jwengle@wisc.edu (J.W.E.); 2Conditions Extrêmes et Matériaux: Haute Température et Irradiation, Centre National de la Recherche Scientifique, UPR3079, Energy, Materials Earth and Universe Science Doctoral School, Université d’Orléans, F-45071 Orléans, France; 3Department of Radiology, University of Wisconsin School of Medicine and Public Health, 1111 Highland Avenue, Madison, WI 53705, USA

**Keywords:** targeted radionuclide therapy, auger emission, radionuclide production, lanthanide separation, erbium-165, ^165^Er

## Abstract

**Background:** Radionuclides emitting Auger electrons (AEs) with low (0.02–50 keV) energy, short (0.0007–40 µm) range, and high (1–10 keV/µm) linear energy transfer may have an important role in the targeted radionuclide therapy of metastatic and disseminated disease. Erbium-165 is a pure AE-emitting radionuclide that is chemically matched to clinical therapeutic radionuclide ^177^Lu, making it a useful tool for fundamental studies on the biological effects of AEs. This work develops new biomedical cyclotron irradiation and radiochemical isolation methods to produce ^165^Er suitable for targeted radionuclide therapeutic studies and characterizes a new such agent targeting prostate-specific membrane antigen. **Methods:** Biomedical cyclotrons proton-irradiated spot-welded Ho_(m)_ targets to produce ^165^Er, which was isolated via cation exchange chromatography (AG 50W-X8, 200–400 mesh, 20 mL) using alpha-hydroxyisobutyrate (70 mM, pH 4.7) followed by LN2 (20–50 µm, 1.3 mL) and bDGA (50–100 µm, 0.2 mL) extraction chromatography. The purified ^165^Er was radiolabeled with standard radiometal chelators and used to produce and characterize a new AE-emitting radiopharmaceutical, [^165^Er]PSMA-617. **Results:** Irradiation of 80–180 mg ^nat^Ho targets with 40 µA of 11–12.5 MeV protons produced ^165^Er at 20–30 MBq·µA^−1^·h^−1^. The 4.9 ± 0.7 h radiochemical isolation yielded ^165^Er in 0.01 M HCl (400 µL) with decay-corrected (DC) yield of 64 ± 2% and a Ho/^165^Er separation factor of (2.8 ± 1.1) · 10^5^. Radiolabeling experiments synthesized [^165^Er]PSMA-617 at DC molar activities of 37–130 GBq·µmol^−1^. **Conclusions:** A 2 h biomedical cyclotron irradiation and 5 h radiochemical separation produced GBq-scale ^165^Er suitable for producing radiopharmaceuticals at molar activities satisfactory for investigations of targeted radionuclide therapeutics. This will enable fundamental radiation biology experiments of pure AE-emitting therapeutic radiopharmaceuticals such as [^165^Er]PSMA-617, which will be used to understand the impact of AEs in PSMA-targeted radionuclide therapy of prostate cancer.

## 1. Introduction

The recent phase III clinical trial of Lutathera^®^ ([^177^Lu]DOTATATE) for neuroendocrine tumors [1] and phase II clinical trial of [^177^Lu]PSMA-617 for prostate cancer [2] show receptor targeted, medium energy electron-emitting radiopharmaceuticals are effective in treating these solid tumors. However, for the treatment of micrometastatic or disseminated cancers, radiopharmaceuticals emitting shorter range, higher linear energy transfer (LET) radiations, such as Auger electrons (AEs, also known as Auger–Meitner or Meitner–Auger electrons [3,4]), show potential in preclinical studies [5,6,7]. Following decay, AE-emitting radionuclides release a cascade of 0.02–50 keV electrons that stop with high LET over subcellular, nano- to micrometer ranges. These properties give AE-based radiopharmaceuticals reduced crossfire irradiation of healthy tissues and off-target dose burden compared with β^−^-emitting radionuclide therapeutics. Furthermore, when AE-emitting radiopharmaceuticals are specifically localized to radiation-sensitive subcellular structures such as DNA [8], this may result in higher maximum tolerated doses and expanded therapeutic windows. The biological impact of these AEs is also evident in comparison studies of ^177^Lu- and ^161^Tb-based radiopharmaceuticals [9,10,11], which are biologically, chemically, and physically matched, with the exception that ^161^Tb has ten times larger AE emission yields (Table 1) [12]. Fundamental radiation biology studies of the effects of AEs require a pure AE-emitting radionuclide, with minimal concomitant medium/high energy electron or photon emissions. Erbium-165 decays by electron capture with only low energy X-ray and AE emissions (Table 1) [12]. As a heavy lanthanide, ^165^Er can radiolabel the same biological targeting vectors used to deliver ^177^Lu and ^161^Tb [13], allowing for comparative studies of pure AE-emitting (^165^Er), β^−^-emitting (^177^Lu), and mixed AE- and β^−^-emitting (^161^Tb) radiopharmaceuticals. Thus, ^165^Er is a useful tool for fundamental studies on the biological effects of AEs and, if incorporated into an appropriate biological targeting vector, for the targeted radionuclide therapy of metastatic and disseminated disease.

Proton, deuteron, or alpha particle irradiation of erbium or holmium targets produces no-carrier-added ^165^Er through a variety of nuclear reaction routes [14]. Proton or deuteron irradiation of erbium produces ^165^Tm (*t*_1/2_ = 30.06 h) via ^nat^Er(*p,xn*)^165^Tm [15,16] or ^nat^Er(*d,xn*)^165^Tm [17], respectively, which can be chemically isolated prior to its β^−^ decay to ^165^Er. While these routes offer high ^165^Er yields, they require a medium energy, multiparticle research cyclotron and expensive, isotopically enriched targets for highest yields. Irradiation of naturally monoisotopic holmium targets produces ^165^Er using 35–70 MeV alpha particles via ^165^Ho(*α,4n*)^165^Tm(β^–^ decay)^165^Er [18,19], 10–20 MeV deuterons via ^165^Ho(*d,2n*)^165^Er [20,21], or 6–16 MeV protons via ^165^Ho(*p,n*)^165^Er [22,23,24]. The existence of more than 500 biomedical cyclotrons capable of accelerating low energy protons makes the latter route particularly accessible to the research community worldwide [25].

Receptor-targeted therapeutic radiopharmaceuticals for human use are radiolabeled using 10–40 GBq ^177^LuCl_3_ with ~90% radiolabeling yield at ~60 MBq/nmol molar activity [26,27,28]. To accomplish these molar activities for biomedical cyclotron-produced ^165^Er, the radionuclide must be chemically isolated from the holmium target material with a high (>10^5^) separation factor (SF) due to residual target material competition with ^165^Er during radiolabeling chemistry. Radiochemical separation of ^165^Er from a Ho cyclotron target is challenging as adjacent lanthanides have identical oxidation states, similar coordination chemistry, and unfavorable ^165^Er (~10 ng) to Ho (~100 mg) mass ratio. Radiochemical separations of adjacent lanthanides have traditionally been performed using cation exchange (CX) chromatography with the complexing agent α-hydroxy isobutyrate (αHIB) [29,30], with recent reports effectively separating Gd/Tb [31,32], Dy/Ho [33], Ho/Er [22,24], and Er/Tm [34,35]. Additionally, extraction chromatography (EXC) resins [36,37] impregnated with acidic organophosphorous extractants including bis(2-ethylhexyl) phosphoric acid (HDEHP, LN resin) [38,39], 2-ethylhexyl phosphonic acid mono-2-ethylhexyl ester (HEHEHP, LN2 resin) [40,41,42,43], and bis(2,4,4-trimethyl-1-pentyl)phosphinic acid (H[TMPeP], LN3 resin) [24,43] have been used to separate adjacent heavy lanthanides.

A recent publication of the biomedical cyclotron production and radiochemical isolation of ^165^Er from holmium reported the production of 1.6 GBq ^165^Er in a 10 h, 10 µA proton irradiation [24]. The ^165^Er was isolated in a 10 h process through a CX/αHIB column using a proprietary resin with 12–22 µm particle size followed by an EXC column using LN3 resin to concentrate the product. While the isolation of ^165^Er from macroscopic Ho was achieved, neither the residual mass of Ho in the ^165^Er product, the Ho/Er SF, nor radiopharmaceutical labeling results were reported [24]. The present work aims to improve the irradiation intensity tolerance of holmium targets to allow for the production of GBq-scale ^165^Er in shorter irradiations, develop a radiochemical isolation process that utilizes commercially available resins to achieve a high Ho/Er SF in shorter times, and demonstrate the radiopharmaceutical quality of the produced ^165^Er through chelator-based titration apparent molar activity (AMA) measurements and labeling a clinically relevant DOTA-based radiopharmaceutical (PSMA-617).

## 2. Materials and Methods

### 2.1. Ho Target Preparation and Irradiation

Cyclotron targets were prepared from holmium metal foils. Initially, 300–640 µm thick holmium foils (99.9%, Alfa Aesar, Haverhill, MA, USA) were used. Based on certificates of analysis, two differing foil lots contained 0.06% (600 ppm) or <0.01% (<100 ppm) erbium. High-purity 0.5 mm-thick, 10 mm diameter holmium metal discs with 0.5 ppm Er were purchased from the U.S. Department of Energy (DOE) Ames Laboratory Materials Preparation Center (MPC), Ames, IA, USA. Based on proton energy loss calculations [44], a 300 µm holmium degrades 12.5 MeV protons to 7.1 MeV, below which the ^165^Ho(*p,n*)^165^Er nuclear reaction cross-section vanishes [24]. Holmium foils were wrapped in 25 µm stainless steel foil and rolled to desired thickness (190–400 µm) using a commercial rolling mill. A commercial disc cutter (Pepetools) was used to punch holmium discs of desired diameter (3–9.5 mm). The holmium target disc was centered and spot-welded to a 0.5 mm-thick, 19 mm diameter tantalum disc using a variable transformer-controlled (60–75% power) commercial 115 V spot welder fitted with a copper or silver electrode as previously described [45]. Roughly 6–10 individual spot welds uniformly cover the 7 to 70 mm^2^ holmium target.

Spot-welded holmium metal targets were proton irradiated at the University of Wisconsin using two biomedical cyclotrons—an RDS-112 (CTI Cyclotron Systems, Knoxville, TN, USA) and a PETtrace (GE Healthcare, Uppsala, Sweden). For RDS-112 irradiations, the target disc was clamped to a water-jet-cooled target support fixture with a 12.7 mm apertured aluminum ring and irradiated with 10–20 µA of undegraded 11 MeV protons. For PETtrace irradiations, a commercial solid target irradiation and transfer system (ARTMS QIS, Vancouver, BC, Canada) was used. Holmium targets discs were assembled into the water-jet cooled transfer capsule, positioned 3.6 cm down-beam from a water-cooled 500 µm thick aluminum degrader, and irradiated with 20–40 µA of 12.5 MeV protons. 

For low intensity irradiations, ^165^Er was quantified by high-purity germanium (HPGe) gamma spectrometry (full width at half maximum at 1333 keV = 1.6 keV, Canberra Inc., Meriden, CT, USA) of the irradiated target disc, while correcting for the self-attenuation of the 46–55 keV x-rays. The HPGe low energy range (30–300 keV) was efficiency calibrated using ^133^Ba and ^241^Am calibration standards (Amersham, United Kingdom).

### 2.2. ^165^Er Radiochemical Isolation

After irradiation, holmium cyclotron targets were dissolved in 11 M HCl (2 mL, Trace Select Ultra, Fluka Analytical, Buchs, Switzerland), followed by evaporation to dryness at 120 °C under Ar flow. The resulting yellow/pink salts were dissolved in 70 mM αHIB (2 mL, pH = 4.7). The αHIB solution was freshly prepared by dissolving α-hydroxyisobutyric acid (99%, Sigma Aldrich, St. Louis, MO, USA or 98%, Acros Organics, Geel, Belgium) with 18 MΩ·cm ultrapure water (Milli-Q) and pH adjusted with 25% ammonia solution (Fisher Scientific, Waltham, MA, USA) using a pH probe (Checker^®^ pH tester, Hanna Instruments, Woonsocket, RI, USA). A 30–60 µL aliquot of the dissolved, reconstituted target was assayed for ^165^Er radioactivity by ionization chamber dose calibrator (CRC-15R, setting #260, Capintec Inc., Florham Park, NJ, USA). The dose calibrator setting #260 was experimentally determined by cross-calibration with HPGe gamma spectrometry. The ^165^Er activity in highly concentrated (50–90 mg/mL) holmium solutions was corrected for self-attenuation of the 46–55 keV X-rays.

Preparation of radiopharmaceutical quality ^165^Er from bulk holmium was accomplished through a three-step radiochemical isolation process. First, a 1 cm diameter, 25 cm-long CX column (AG50W-X8, 230–400 mesh, 63–150 µm, 1.7 meq/mL, NH_4_^+^ form, Bio-Rad, Hercules, CA, USA) was equilibrated with water (~100 mL), then 70 mM αHIB (pH = 4.7, ~100 mL), followed by injection of the ^165^Er/Ho/αHIB solution and elution with 5 mL/min 70 mM αHIB (pH = 4.7, 440–820 mL). Under these conditions, ^165^Er elutes before holmium and the first ~90% of eluted ^165^Er was collected (150–350 mL) for secondary purification, as determined by monitoring the column effluent using a shielded inline radiation detection system consisting of a CsI(Tl) scintillator (1 × 1 cm, Hilger Crystals Ltd., Kent, UK) coupled to a photomultiplier tube (E849-35, Hamamatsu Photonics, Hamamatsu City, Japan) powered and processed with bench-top electronics (925-SCINT, Ortec, Oak Ridge, TN, USA) and logged using a digital counter (USB-6008, National Instruments Corp., Austin, TX, USA). Then, the mobile phase was changed to 0.5 M αHIB (pH = 4.7, 250 mL) for stripping the remaining bulk holmium from the cation exchange column followed by water (200–500 mL) for column storage. The ^165^Er radioactivity eluting in each collected CX fraction was quantified by a dose calibrator immediately after elution. 

The second step utilized EXC with a commercially available HEHEHP-impregnated resin (LN2, 20–50 µm, Triskem Int., Bruz, France) [37], based on previous literature work on its use for Ho/Er separations [41]. Fritted polypropylene columns (5.5 mm diameter, 1 mL, Supelco Inc., Bellefonte, PA, USA) were dry-packed with resin (500 mg) and preconditioned with 1 M HNO_3_ (5 mL) followed by 0.1 M HNO_3_ (25 mL). All nitric acid solutions were prepared using 16 M HNO_3_ (TraceSelect, Fluka Analytical, Buchs, Switzerland) and 18 MΩ·cm ultrapure water (Milli-Q). The ^165^Er/Ho/αHIB eluate from separation step 1 was acidified to 0.1 M HNO_3_ using 16 M HNO_3_ and passed through the EXC column at 5.8 ± 0.9 mL/min using a peristaltic pump (*n* = 10, WPM1-P1CA-WP Welco Co. Ltd., Fuchū, Japan). With a lower flow rate of 1.2 ± 0.1 mL/min (*n* = 10, WPM1-P1BB-BP, Welco Co. Ltd., Fuchū, Japan), holmium was eluted with 0.4 M HNO_3_ (40–60 mL), followed by ^165^Er eluted with 1 M HNO_3_ (4–6 mL). 

In the third separation step, the ^165^Er-rich fractions from separation step 2 were acidified from 1 M to 5 M HNO_3_, loaded onto a N,N,N′,N′-tetra-2-ethylhexyldiglycolamide (100 mg, bDGA, 50–100 µm, Triskem Int., Bruz, France) EXC column. The column was then rinsed with 3 M HNO_3_ (15 mL) and 0.5 M HNO_3_ (2 mL). The ^165^Er was subsequently eluted with 0.01 M HCl (1–1.5 mL).

### 2.3. ^165^Er Quality Control

Holmium mass across the separation procedure was quantified using microwave plasma atomic emission spectrometry (MP-AES, MP4200, Agilent Technologies, Santa Clara, CA, USA). Holmium standard solutions of 0.1–50 ppm were made by dissolving holmium chloride hydrate (REaction^®^ 99.99% (REO), Alfa Aesar, Haverhill, MA, USA) in water or holmium metal (US DOE Ames Laboratory MPC, Ames, IA, USA) in 11 M HCl, followed by dilution in 0.1 M HCl. The limit of detection for holmium was determined to be ~0.1 ppm in an undiluted sample solution. The MP-AES-analyzed samples were typically diluted by a factor of 2–10 in 0.1 M HCl. The Ho/Er SFs of the various separation steps and the overall separation procedure were calculated according to Equation (1) with *m_Ho,before/after_* being the MP-AES-quantified holmium mass before/after the separation step and *A_^165^Er, before/after_* being the dose-calibrator-quantified ^165^Er activity before/after the separation step.
(1)SFHo/Er=mHo, beforemHo,afterA165Er,beforeA165Er,after

The apparent molar activity (AMA) of the final ^165^Er solution was determined by titration using tetraazacyclododecane-1,4,7,10-tetraacetic acid (DOTA, Macrocyclics Inc, Plano, TX, USA) and diethylenetriaminepentaacetic acid (DTPA, Acros Organics, Geel, Belgium) as previously described [46]. To polypropylene tubes, 0.5–3 MBq (20 µL) of ^165^Er in 0.01 M HCl, 1 M NaOAc (100 µL, pH 4.7, 99.995% trace metals basis, Aldrich, St. Louis, MO), and DOTA or DTPA (100 µL, 0.03–3 µg/mL) in 18 MΩ·cm water were added. Following incubation at 85 °C for DOTA and 21 °C for DTPA for 1 h, each tube was assayed by thin layer chromatography (TLC) using silica-based stationary phase (J.T. Baker, Phillipsburg, NJ, USA) and 0.05 M disodium ethylenediaminetetraacetic acid (EDTA, Fisher Scientific Co., Pittsburgh, PA, USA) as mobile phase. Radioactivity distribution on the TLC plates was visualized using a Cyclone Plus phosphor plate reader (Perkin Elmer, Waltham, MA, USA). Free ^165^Er had a retention factor (R_f_) of ~1, chelated ^165^Er-DOTA had an R_f_ of ~0.2, and ^165^Er-DTPA had an R_f_ of ~0.7. To compute the AMA, ^165^Er activity in MBq was divided by twice the number of moles of DOTA/DTPA required to complex 50% of the radioactivity, and the value was reported as MBq of ^165^Er per nmol of ligand, or MBq/nmol (mean ± standard deviation, SD).

### 2.4. Radiosynthesis and Characterization of [^165^Er]PSMA-617

PSMA-617 (MedChemExpress, Monmouth Junction, NJ, USA) was dissolved in 18 MΩ·cm water to a concentration of 0.1 µg/µL and distributed into 10 aliquots that were stored at –20 °C. A sodium acetate solution (1 M, 99.995% trace metals basis, Aldrich) was buffered with hydrochloric acid until a pH of 5.7 was obtained. Er-165 (41–52 MBq in 70–80 µL 0.01 M HCl) was added to a solution of PSMA-617 in water (1 nmol in 10 µL, 2 nmol in 10 µL, or 5 nmol in 25 µL), NaOAc aq. (1M, pH 5.7, 50 µL), and L-ascorbic acid (0.3–0.4 mg, 25 µL, ≥99.9998 trace metals basis, Honeywell-Fluka), which were reacted at 80 °C and 1000 rpm for 30 min. The reaction was diluted in 18 MΩ·cm water (10 mL) and loaded onto a pre-equilibrated C-18 Plus Light cartridge; after rinsing with water (10 mL), elution was performed with pure ethanol (700 µL). An aliquot (25 µL) was diluted with 18 MΩ·cm water (25 µL) and set aside for quality control analysis. Analytical HPLC was performed on [^165^Er]PSMA-617 using an Agilent 1260 Infinity II module coupled with an inline radioactivity detection system similar to that described in Section 2.2 but with electronic analog pulse converted to voltage (Model 106, Lawson Labs Inc., Malvern, PA, USA) and logged using an Agilent 1200 universal interface box. The separation was performed on reverse-phase C18 column (InfinityLab Poroshell 120 EC-C18, 4.6 × 100 mm, 2.7 µm, Agilent Technologies, Santa Clara, CA, USA) using a linear gradient (95 to 20% over 15 min) of 0.1% trifluoroacetic acid (Thermo Scientific in 18 MΩ·cm water) in acetonitrile (HPLC-grade, Fisher Scientific, Pittsburgh, PA, USA) at 1 mL/min.

In vitro stability of [^165^Er]PSMA-617 was investigated in the presence of L-ascorbic acid and freshly prepared normal human serum prepared from lyophilized powder (009-000-001, Jackson ImmunoResearch Laboratories, Inc., West Grove, PA, USA) reconstituted using phosphate buffered saline (2.0 mL, PBS, Lonza Bioscience). Dry [^165^Er]PSMA-617 (3.1 MBq) and L-ascorbic acid (0.6 mg) were dissolved in serum (300 µL) and incubated at 37 °C for 12 h. Aliquots of the [^165^Er]PSMA-617 complex were assessed by analytical HPLC at t = 1 and 12 h. The chromatograms were analyzed by integration of the [^165^Er]PSMA-617 peak compared to all radioactive peaks (free erbium or decomposition products) in the chromatogram. 

The distribution coefficient (logD value) of [^165^Er]PSMA-617 was determined using a 1:1 (*v*/*v*) solution of n-octanol (Alfa Aesar, Haverhill, MA, USA) and PBS according to previously reported methods [11]. A sample of [^165^Er]PSMA-617 (8.2–51 MBq, 2.8–8.4 MBq/nmol) was dried and diluted with PBS and n-octanol (700 µL each). The solution was vigorously agitated for 5 min before being centrifuged at 1000 rpm for 5 min. An aliquot of PBS and n-octanol was analyzed by HPGe gamma spectrometry under identical geometries and the ratio of decay-corrected net counts per minute was used to determine the distribution coefficient. The PBS aliquot required 4–5 days of decay before acquisition of an HPGe spectrum with acceptable (<5%) dead-time. Uncertainty in the LogD value was calculated by propagation of error associated with HPGe counting statistics and in the half-life of ^165^Er (10.36 ± 0.04 h [47]). The logD experiment was repeated for five independent preparations of [^165^Er]PSMA-617 and logD reported as average and standard deviation of results.

## 3. Results

### 3.1. Ho Target Preparation and Irradiation

Cold rolling, disc cutting, and spot-welding methods were well suited for the fabrication of cyclotron irradiation targets of tightly controlled dimensions and mass from a variety of commercial holmium metal sources. The malleability of holmium allowed for the dramatic thinning of metal foils through rolling. While thickness reduction by factors of two or three was readily achieved, when an eight-fold change in thickness was attempted, cracking around the edges was observed, as shown in supplementary Appendix A. Spot-welded holmium was well adhered to the tantalum backing and withstood proton irradiation at all investigated intensities with only minor discoloration, as shown in Figure 1.

For the PETtrace cyclotron, a proton irradiation energy of 12.5 MeV centers the ^165^Ho(*p,n*)^165^Er excitation function peak (see Appendix A [22,23,24]) within the energy loss window of the protons traversing a 200–300 µm-thick holmium target. Experimental end-of-bombardment (EoB) ^165^Er physical yields were measured via attenuation-corrected HPGe of target discs or dose calibrator measurements of dissolved target aliquots, or CX elution fractions (Table 2). The yields show a significant dependence on the holmium target dimensions and the irradiating cyclotron. Calculated using literature cross-sections [23,24], a 1–2 h proton irradiation of 275 µm-thick holmium results in physical ^165^Er yields of 50 MBq·µA^−1^·h^−1^ at 12.5 MeV and 39 MBq·µA^−1^·h^−1^ at 11 MeV. Experimental ^165^Er physical yields were significantly lower than these theoretical maxima, likely because the cyclotron-integrated charge includes protons impinging on the target outside the holmium diameter. This is especially problematic for the PETtrace cyclotron, which has an oblong beam spot with full width at half maxima of 11 and 8.7 mm, as measured by autoradiography of irradiated aluminum discs. The RDS-112 cyclotron provides a significantly smaller beam spot, resulting in higher overall ^165^Er physical yield, despite the lower irradiation energy and smaller target diameter. However, the RDS-112 cyclotron is limited to maximum irradiation current of 20 µA, a factor of 2 below that routinely used with the PETtrace cyclotron.

### 3.2. ^165^Er Radiochemical Isolation

The holmium target was dissolved over 5 min at room temperature and evaporated to dryness in 30 min. The time of dissolution/evaporation/reconstitution/CX injection was 50 ± 20 min (*n* = 13).

Step 1: CX/αHIB

The first step of the ^165^Er isolation procedure accomplishes a bulk holmium/erbium separation while accommodating 180 mg holmium loading masses through CX/αHIB column chromatography with commercially available CX resin in a standard stainless steel semipreparative high pressure chromatography column housing. This 19.6 mL column has a theoretical capacity of 1.8 g of trivalent Ho^3+^, ten times larger than the intended holmium loading masses. Based on the Dy/Ho separation process of Mocko et al. [33], 70 mM αHIB (pH = 4.7) was used as mobile phase. When mobile phase was freshly prepared and carefully pH adjusted to within 0.05 pH units, consistent retention times were observed with 90% of the total ^165^Er radioactivity eluting in a Gaussian-shaped peak 40–90 min after injection, as shown in the representative radiochromatogram (Figure 2). Holmium elutes after ^165^Er with its leading edge beginning at ~350 mL as seen through the diminishing Ho/Er SF with increasing ^165^Er fraction collection volume. Use of mobile phase that was not freshly prepared or incorrectly adjusted to too low of a pH resulted in significantly longer retention times and diminished separation of ^165^Er and Ho. With 5 mL/min flow, the column pressure was routinely 17–19 MPa. Following ^165^Er elution, the column was stripped with freshly prepared 0.5 M αHIB (100 mL, pH = 4.7), followed by water (250 mL). The column was re-used until the flow pressure significantly increased (>22 MPa), upon which it was disassembled and repacked with fresh resin slurry, every ~10 uses.

As determined by dose calibrator measurements of ^165^Er and MP-AES measurements of Ho in the eluted CX fractions, the CX/αHIB column accommodated 180 mg Ho loading mass and effectively removed bulk Ho with acceptable ^165^Er yield. Loading 111 ± 17 mg Ho and recovering 94.7 ± 2.5 % of ^165^Er resulted in a SF of 320 ± 210 (*n* = 6). Loading 174 ± 8 mg Ho and recovering 95.3 ± 1.8 % of ^165^Er resulted in a SF of 130 ± 60 (*n* = 5). For the larger loading masses, decreasing ^165^Er recovery to 90.5 ± 1.4 % resulted in an SF of 250 ± 150 (*n* = 5) and further decreasing ^165^Er recovery to 80% resulted in a SF of 1000 ± 400 (*n* = 1). These results indicate the sensitivity of the CX/αHIB separation to Ho loading mass and demonstrate the challenging balance between ^165^Er recovery and Ho/Er SF. To ensure a reproducible, optimal balance between yield and SF for this step, an inline radiation detector was used to determine when to stop collecting the ^165^Er fraction. Following loading ≤120 mg Ho, ^165^Er fraction collection was ended when the radioactivity signal was ~1/10th maximum value, resulting in ~95% ^165^Er recovery. Following loading ~180 mg Ho, ^165^Er fraction collection was ended when the radioactivity signal was ~1/4th maximum value, resulting in ~90% ^165^Er recovery.

Step 2: LN2 EXC

The second step of the ^165^Er isolation procedure accomplishes a high Ho/Er SF while accommodating milligram quantity holmium masses through EXC using commercially available LN2 resin in a polypropylene column. When filled to maximum capacity (500 mg), the 1.3 mL column has a theoretical capacity of 36 mg of trivalent lanthanides according to the Triskem product sheet. However, a significant decrease in chromatographic performance was observed when loading more than 5% theoretical capacity, limiting the LN2 column capacity to 1–2 milligrams of holmium (see Appendix A). Following the CX/αHIB column, the acidified Ho/^165^Er solution was loaded onto the LN2 column, trapping both ^165^Er and Ho. Based on previously published studies [41], the column was rinsed with 0.4 M HNO_3_ to affect the differential elution of holmium before erbium. As shown in Figure 3, elution with 0.4 M HNO_3_ (50 mL) removed >99% of the holmium, along with a cumulated ~20% of the ^165^Er. The remaining ^165^Er was rapidly eluted with 1 M HNO_3_ (~5 mL). To avoid moderate to severe decrease in chromatographic performance, care was taken to prevent the resin bed from going dry during use and columns were freshly packed and conditioned prior to each experiment (see Appendix A).

In the optimized procedure, LN2 columns loaded with Ho (570 ± 370 µg) and rinsed with 0.4 M HNO_3_ (52 ± 9 mL) resulted in 78 ± 6% ^165^Er recovery and a Ho/Er SF of 1020 ± 320 (*n* = 4). The LN2 SF was estimated from the Ho mass in the final preparation of ^165^Er, assuming no Ho/Er separation was achieved with the final bDGA column. Three additional replicates where holmium was below the MP-AES detection limit and four additional replicates with deviations from the above-described experimental procedure were excluded from this analysis. In two of these excluded replicates, a 0.4 M HNO_3_ (42 mL) rinse resulted in 96% ^165^Er recovery and a Ho/Er SF of 290 and a 0.4 M HNO_3_ (52 mL) rinse resulted in a 63% ^165^Er recovery and an SF of >2000. These results demonstrate the sensitivity of the procedure to HNO_3_ concentration/volume and the interplay between Ho/Er SF and ^165^Er recovery. The latter of the two results indicates that simply using a set volume of 0.4 M HNO_3_ to remove Ho may lead to irreproducible ^165^Er recovery and Ho/Er SF. To ensure a reproducible, optimal balance between recovery and SF, the radioactivity in the 0.4 M HNO_3_ eluant is quantified by a dose calibrator as it is collected in 1–5 mL fractions. After ~20% of the loaded radioactivity had eluted, the mobile phase was changed to 1 M HNO_3_ to elute the remaining high purity ^165^Er with the optimized results reported above. 

Step 3: bDGA EXC

The final step of the ^165^Er isolation procedure reduced trace metal impurities while concentrating the product into a small volume, low acidity solution suitable for radiolabeling using a commercially available bDGA resin. The loading (5 M HNO_3,_ 5 mL) and rinsing (3 M HNO_3,_ 15 mL) solution concentrations were chosen due to literature studies showing high erbium and low transition metal (Fe, Co, Ni, Cu, Zn) affinities on a similar DGA resin [48]. A small volume rinse (0.5 M HNO_3_, 2 mL) decreased column acidity and minimized subsequent ^165^Er elution volume. Following this loading and rinsing routine, 97 ± 2 % of the loaded ^165^Er was recovered in 0.01 M HCl (1.2 ± 0.2 mL) (*n* = 18). Smaller elution volumes were attainable by fractionation of the 0.01 M HCl elution solution, with the optimized (*n* = 5) elution profile shown in Table 3.

Overall separation:

The overall chemical isolation procedure from start-of-dissolution to final ^165^Er isolation in 0.01 M HCl (~400 µL) was 4.9 ± 0.7 h (*n* = 13). The optimized procedure had a decay-corrected ^165^Er recovery of 64 ± 2% and a Ho/Er SF of (2.8 ± 1.1)·10^5^ and was validated for holmium target masses up to 180 mg, with the final ^165^Er fraction containing 370 ± 180 ng of residual holmium and no detectable radionuclidic impurities (Appendix A) after separation (*n* = 4).

### 3.3. DTPA/DOTA AMA Determination

Following successful ^165^Er isolation from low-purity Ho foils (<100 ppm Er, Alfa Aesar, Haverhill, MA, USA) and high-purity Ho foils (0.5 ppm Er, DOE Ames Laboratory MPC, Ames, IA, USA), EoB-decay-corrected AMA values determined by DTPA and DOTA titration are shown in Table 4.

### 3.4. Radiosynthesis and Characterization of [^165^Er]PSMA-617

Following successful isolation, the [^165^Er]PSMA-617 was synthesized with the labeling yields summarized in Table 5. Radioanalytical HPLC showed [^165^Er]PSMA-617 (UV retention time (RT) = 6.45 min, radioactivity RT = 6.60), co-eluting with cold Er-PSMA-617 and Ho-PSMA-617 standards (230 nm RT = 6.45 min), shown in Appendix A. Radiochemical purity of the [^165^Er]PSMA-617 was > 95%. Visible also in the 230 nm absorbance chromatograph of the [^165^Er]PSMA-617 radiopharmaceutical were multiple mass peaks with RT = 6.8–7.0 min, retention times corresponding to [^nat^Zn]PSMA-617 (230 nm RT = 6.90 min), [^nat^Fe]PSMA-617 (230 nm RT = 6.80 min), and [^nat^Cu]PSMA-617 (230 nm RT = 6.95 min) (see Appendix A). The [^165^Er]PSMA-617 radiochemical purity remained > 95% after 12 h incubation in whole human serum, the longest time point assessed. The octanol-PBS distribution coefficient of [^165^Er]PSMA-617 was −3.3 ± 0.3 (*n* = 5).

## 4. Discussion

For application in receptor-targeted therapeutic radiopharmaceuticals, a high molar activity is necessary to achieve good radiolabeling yield of small pharmaceutical masses with large amounts of radioactivity. This high molar activity ensures that the biologically administered mass of radiopharmaceutical is sufficiently small to not saturate the targeted receptor on diseased cells. When prepared for human use, [^177^Lu]PSMA-617 and [^177^Lu]DOTATATE are radiolabeled with high yield at a molar activity of 60 MBq/nmol [26,27]. For cyclotron-produced ^165^Er, achieving a high molar activity must begin with careful consideration of the holmium target material. Many commercial sources of holmium have significant erbium impurity, which, in turn, will limit the maximum attainable molar activity of ^165^Er produced in the proton irradiation of an impure holmium target. Additionally, as shown in Table 2, the holmium target and irradiation parameters significantly affect the overall quantity of ^165^Er that can be produced in a bombardment, with larger targets having greater overall yields compared to their smaller counterparts. However, larger holmium mass targets come with the added challenges of a more difficult ^165^Er/Ho separation and a higher cold erbium and holmium burden in the purified ^165^Er fraction. 

In addition to sourcing Ho target material with low Er impurity content, an effective Ho/Er radiochemical isolation procedure is necessary. Because of the chemical similarity between the adjacent lanthanide elements, any residual holmium in the ^165^Er final formulation will affect the molar activity of ^165^Er. A high Ho/Er SF is accomplished in this work by a multi-step separation process utilizing cation exchange and extraction chromatography. Based on MP-AES analysis of the final ^165^Er product, this ^165^Er radiochemical isolation process gives a Ho/Er SF of (2.8 ± 1.1) · 10^5^. The ^165^Er radiochemical yield was 64 ± 2% calculated by the ratio of the decay-corrected ^165^Er activity in the final product and the ^165^Er activity produced in the irradiated target.

Two main sources of erbium/holmium—cold erbium target impurity and residual holmium due to incomplete separation—are the two dominating terms in the denominator of Appendix A, which calculates a maximum achievable EoB MA for a given ^165^Er preparation. With the experimental ^165^Er production and radiochemical separation results presented above, Appendix A yields a calculated EoB ^165^Er MA of 9.9 ± 0.5 MBq/nmol for a 1 h, 12.5 MeV, 130 mg, 9.5 mm Ø, low purity (100 ppm Er) holmium target PETtrace irradiation. This calculated MA_EoB_ is in good agreement with the measured DOTA/DTPA ^165^Er AMAs (Table 4). In this case, the calculated EoB MA is nearly entirely driven by the cold erbium impurity present in the holmium target material (second term in Appendix A denominator). This underscores the fact that under the investigated irradiation conditions, utilization of holmium targets with high erbium impurity content will limit the molar activity of the resulting ^165^Er to values lower than typically acceptable for therapeutic radiopharmaceutical research applications. Producing high MA ^165^Er requires holmium with extremely low erbium content, such as target material that has been prepurified from erbium, or the DOE Ames Laboratory MPC metal used in this work.

For an identical irradiation of a high-purity (0.5 ppm Er) holmium target, Appendix A yields an ^165^Er EoB MA of 240 ± 60 MBq/nmol, with this molar activity being driven by cold holmium remaining in the ^165^Er preparation after their separation by a factor of (2.8 ± 1.1) · 10^5^ (third term in Appendix A denominator). However, the EoB ^165^Er AMAs (Table 4) with DOTA (1.2–47 GBq/µmol_DOTA_, *n* = 3) and DTPA (4.9–250 GBq/µmol_DTPA_, *n* = 5) and radiolabeled [^165^Er]PSMA-617 MAs (37–130 GBq/µmol_PSMA-617_, *n* = 5, Table 5) do not reflect this high value of MA. This disagreement may be a result of nanomole, sub-ppm trace metal (Zn, Fe, Cu) impurities in the radiolabeling reactions, which are not considered in Appendix A. This hypothesis is supported by the fact that, for ^165^Er isolated from high-purity holmium targets, the titration-based AMA measured using DTPA were 5–22 times higher than for DOTA (*n* = 3). This difference in DTPA/DOTA AMA values, which has also been observed for cyclotron-produced ^86^Y [49], is likely due to the non-selective nature of DOTA’s metal binding properties. Compared with DTPA, DOTA binds 10,000 times stronger to Fe^2+^ (LogK_FeDOTA_ = 20.22 ± 0.07 [50], LogK_FeDTPA_ = 16.0 ± 0.1 [51]), 100 times stronger to Zn^2+^ (LogK_ZnDOTA_ = 20.8 ± 0.2, LogK_ZnDTPA_ = 18.6 ± 0.1 [51]), and 10 times stronger to Cu^2+^ (LogK_CuDOTA_ = 22.3 ± 0.1, LogK_CuDTPA_ = 21.5 ± 0.1 [51]), causing these three common trace metal impurities to be significantly more problematic in DOTA versus DTPA radiochemical labelings. Thus, a systematically higher DTPA-based AMA compared with DOTA-based AMA is indicative that Fe/Zn/Cu-based trace metal impurities in the radiolabeling solutions are significantly impacting the ^165^Er AMA values. The impact of trace Zn, Fe, and Cu on the [^165^Er]PSMA-617 radiolabeling experiments is supported by the presence of UV-absorbing impurities with analytical HPLC retention times equivalent to ^nat^Zn-PSMA-617, ^nat^Fe-PSMA-617, and ^nat^Cu-PSMA-617 in the final radiopharmaceutical preparation (Appendix A). 

This work represents the first published radiosynthesis of [^165^Er]PSMA-617, a radiopharmaceutical that could serve as a useful in vitro and in vivo tool that can be used to assess the role of AEs in the efficacy of PSMA-targeted radionuclide therapy of prostate cancer using [^161^Tb]PSMA-617 [11]. The radiopharmaceutical is stable in serum for at least 12 h and has an octanol–water partition coefficient of LogD = −3.3 ± 0.3, less polar than [^161^Tb]PSMA-617 (−3.9 ± 0.1) [11], [^44^Sc]PSMA-617 (−4.21 ± 0.04), [^177^Lu]PSMA-617 (−4.18 ± 0.06), and [^68^Ga]PSMA-617 (−4.3 ± 0.1) [52].

## 5. Conclusions

A 2 h biomedical cyclotron irradiation and 5 h radiochemical separation can produce GBq-scale ^165^Er suitable for high-yield radiolabelings of DOTA-based radiopharmaceuticals at molar activities befitting investigations of targeted radionuclide therapeutics. The separation utilizes column chromatography with commercially available resins and is well suited for automation. This significant step forward in the production and high holmium/erbium SF radiochemical isolation of ^165^Er will enable fundamental radiation biology experiments of pure AE-emitting therapeutic radiopharmaceuticals. Proof-of-concept radiolabeling studies were successfully performed synthesizing [^165^Er]PSMA-617, which will be utilized in vitro and in vivo to understand the role of AEs in PSMA-targeted radionuclide therapy of prostate cancer.

## Figures and Tables

**Figure 1 molecules-26-07513-f001:**
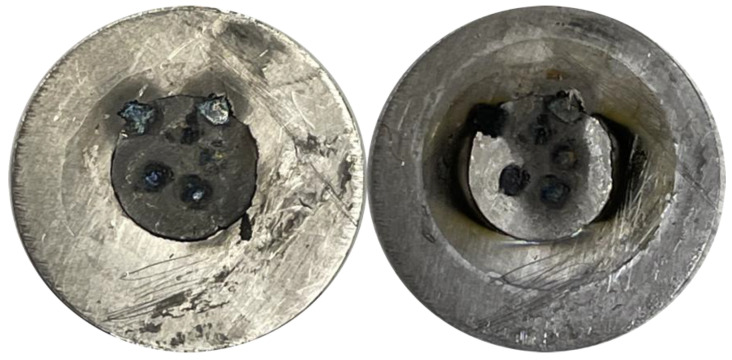
Representative image of a 7.9 mm ø, 270 µm-thick, 106 mg holmium disc spot-welded to tantalum, before and after 68 min, 40 µA·h PETtrace irradiation.

**Figure 2 molecules-26-07513-f002:**
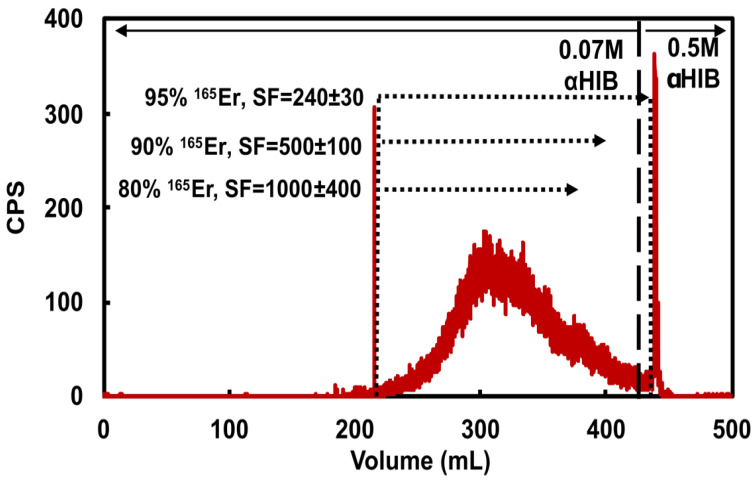
Representative ^165^Er radioactivity elution profile from a cation exchange column loaded with 178 mg holmium and eluted with 5 mL/min 0.07 M αHIB (pH = 4.7). Dotted lines and corresponding text highlight three possible ^165^Er-rich fraction collection volumes demonstrating the balance between ^165^Er recovery and Ho/Er SF. The 0.5 M αHIB (pH = 4.7) rapidly elutes remaining ^165^Er along with bulk holmium.

**Figure 3 molecules-26-07513-f003:**
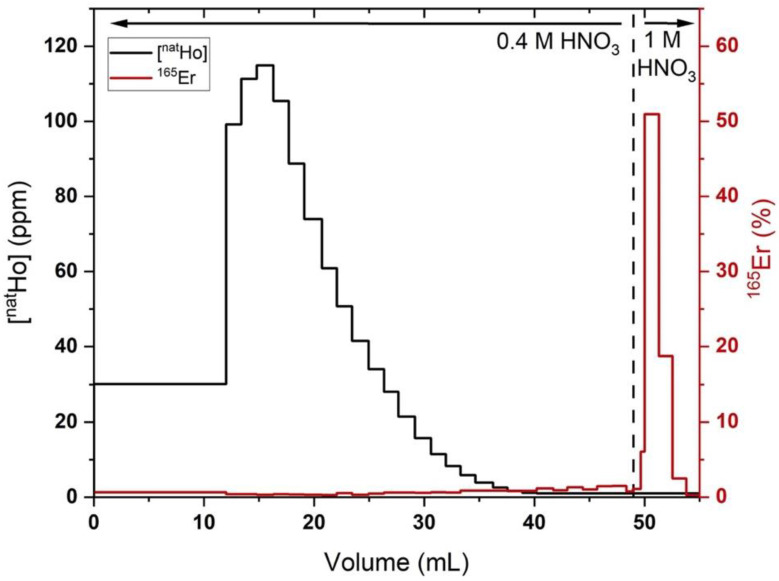
Holmium (black lines, quantified by MP-AES) and ^165^Er (red lines, quantified by radioactivity dose calibrator) elution profiles from a representative 500 mg LN2 column loaded with ~1 mg holmium, ~2 MBq ^165^Er in 0.1 M HNO_3_ (200 mL), 70 mM αHIB and eluted with 0.4 M HNO_3_ (50 mL), followed by 1 M HNO_3_ (5 mL). Fractions with Ho/^165^Er below limits of detection (Ho MP-AES: 1 ppm, ^165^Er: 4 kBq) shown as upper limits.

**Table 1 molecules-26-07513-t001:** Comparison of AEs and β^−^-particles emitted by ^165^Er, ^161^Tb, and ^177^Lu.

R.	Half-Life (d)	Avg. AEs per Decay	Avg. Energy per AE (keV)	Avg. β^−^ per Decay	Avg. Energy per β^−^ (keV)
^177^Lu	6.64	1.1	1	1	133
^161^Tb	6.89	11	5.7	1	154
^165^Er	0.43	7.3	11	0	0

**Table 2 molecules-26-07513-t002:** Erbium-165 physical yields for different Ho targets and irradiation configurations.

Cyclotron	Ho Dimensions			E_in_ (MeV)	E_out_ (MeV)	^165^Er Physical Yield (MBq·µA^-1^·h^-1^)	*n*
	Diam (mm)	Thick. (mm)	Mass (mg)				
PETtrace	9.5	280–300	174 ± 8	12.5	7.5	24.1 ± 0.5	5
PETtrace	9.5	200–240	125 ± 6	12.5	8.4–9.1	19.1 ± 1.1	3
PETtrace	7.9	270–280	108 ± 4	12.5	7.8	14.1 ± 1.4	3
PETtrace	7.9	190	69 ± 1	12.5	9.3	12.0 ± 0.9	4
RDS-112	7.9	190–280	84 ± 22	11	5.3–7.5	28.0 ± 1.8	5
RDS-112	6.4	320–620	121 ± 75	11	<4.8	30.0 ± 6.1	2
RDS-112	4.8	320	48	11	4.2	23	1
RDS-112	4.8	180	23	11	7.7	13	1
RDS-112	3	320	22	11	4.2	16	1
RDS-112	3	180	10	11	7.7	9	1

**Table 3 molecules-26-07513-t003:** Optimized ^165^Er elution profile bDGA EXC column.

Fraction	Volume (µL)	^165^Er Yield (%)
1	210 ± 20	0.1 ± 0.2
2	390 ± 30	88 ± 4
3	420 ± 40	9 ± 4
4	490 ± 30	1.0 ± 0.3
Column	dry	1.8 ± 0.5

**Table 4 molecules-26-07513-t004:** Chelator-titration-based AMA results for ^165^Er isolated from various qualities of Ho targets.

Ho Target Er Impurity (ppm)	DOTA AMA ^†^ (MBq/nmol)	DTPA AMA ^†^ (MBq/nmol)	*n*
<100	11 ± 4	10 ± 4	5
0.5	20 ± 24 *	92 ± 97	4

^†^ AMA decay-corrected to end of bombardment. * *n* = 3 replicates.

**Table 5 molecules-26-07513-t005:** [^165^Er]PSMA-617 radiolabeling yields.

PSMA-617 (nmol)	^165^Er Activity^†^ (MBq)	Labeling Yield (%)	Labeled MA ^†^ (MBq/nmol)	*n*
1	170 ± 88	49 ± 7	82 ± 45	3
2	76	97	37	1
5	250	100	50	1

^†^ Radioactivity and MA decay-corrected to end of bombardment.

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
