# Peer review of "A High Separation Factor for ^165^Er from Ho for Targeted Radionuclide Therapy"

_molecules, 2021, doi:10.3390/molecules26247513_

Round 1
Reviewer 1 Report
The manuscript is very well written, incorporating all the appropriate information required for the readers about cyclotron production procedure (such as Holmium targetry and radiochemical separation of Er from Ho) and radiolabeling of 165Er with PSMA peptide. There are few minor comments (see below) that I would suggest the authors for improving the quality of the manuscript. (→ : replace with)
Title:
- Auger electron therapy → Targeted radionuclide therapy or Targeted auger therapy
Abstract:
- Line 19: Ho(m)→ monoisotopic 165Ho
Keywords:
- Line 34: lanthanide → 165Ho
Introduction:
- Line 48: cross-irradiation → cross-fire irradiation
English Editing:
Please re-check contents of manuscript. I see some errors in your manuscript such as:
Line 19: antigen.Biomedical
Line 41: receptor targeted, medium-energy
The following article (previously made regarding these reactions) should be given as references in the text:
Nuclear Model Calculations on the Production of Auger Emitter 165Er for Targeted Radionuclide Therapy, M Sadeghi, M Enferadi, C Tenreiro, J Mod Phys 4, 217-225
With these small corrections, the manuscript can be published in Molecules Journal. I do not need to see it once more.

Author Response
We would like to thank both reviewers for their constructive criticisms of our work. We have carefully considered their comments and incorporated a number of changes that we feel have significantly strengthened our manuscript. Direct responses to each reviewer comment are found below and prepared the attached modified manuscript and supplementary material documents.
Reviewer 1:
- As suggested by the reviewer, we have changed ‘Auger electron therapy’ to ‘targeted radionuclide therapy’ in the title of both main manuscript and supplementary material.
- We agree with the reviewer that the isotopic make-up of the target material would be a useful addition to the abstract methods section. As a result, we changed ‘…Ho(m)…’ to ‘…naturally monoisotopic 165Ho(m)…’. We maintained the subscripted (m) to convey that the targets were metallic and added ‘naturally’ to avoid the confusion that an isotopically-enriched target material was used.
- The purpose of keywords is to list broader scientific categories to which the work pertains for the purpose of database queries. Because of the chemical similarities of the lanthanide elements, the scientific details of our Ho/Er separation will likely be useful for researchers looking to perform other adjacent lanthanide separations (such as Gd/Tb or Yb/Lu). As such, we believe it is useful to use the broader term ‘lanthanide separations’ as keyword for this work.
- We agree with the reviewer’s suggestion and have changed ‘cross-irradiation’ to ‘cross-fire irradiation’.
- ‘Line 19: antigen.Biomedical’. Our submitted abstract did not have this formatting problem. Perhaps this irregularity occurred because of the automated typesetting of the abstract.
- ‘Line 41: receptor targeted, medium-energy’ We have corrected the hyphenation in this sentence to read, “… receptor targeted, medium energy electron-emitting radiopharmaceuticals…”.
- We have added the suggested reference to Sadeghi et al J Mod Phys 2010 to the first sentence of the second paragraph of the introduction.

Reviewer 2 Report
The scientific work presented in the proposed paper can be considered as an addition to the 2020 PSI publication. Using known methods described for lanthanide separation, the authors propose the production of 165Er the labelling of a PSMA derivative.
Nevertheless before publication additional information has to be added:
-Gamma spectrum of the 165Er after the chemical separation with the radiochemical purity.
- ICP MS of the final 165Er solution with the quantification of all the metals.
- comparison of the obtained values with the titration with DOTA and DPTA and the AMA. Based on the height of the peaks in Supplementary Figure S6 and Supplementary Figure S7 you have 10 times more "Zn" than Er and Ho. Can you confirm these data with ICP MS?
- The UV trace of the HPLC (Supplementary Figure S6)
shows well defined peaks and the gamma trace has such a tailing. Why? - If we want to develop the production of " new radionuclides"around the world we have to propose a possible way to automate the process. Is the proposed process can be automated? If yes how?
Author Response
We would like to thank both reviewers for their constructive criticisms of our work. We have carefully considered their comments and incorporated a number of changes that we feel have significantly strengthened our manuscript. Direct responses to each reviewer comment are found below and prepared the attached modified manuscript and supplementary material documents.
Reviewer 2:
- We agree that a gamma spectrum of the final purified 165Er would be a useful addition to the supplementary material. We have added the following statement to Results section 3b, “…and no detectable radionuclidic impurities (Supplementary Figure S6)…” and added such a spectrum as Supplementary Figure S6. This has resulted in the changing of former Supplementary Figure S6 to S7 and S7 to S8, with changes made to both the manuscript and supplementary materials.
- In their second and third points, the reviewer is requesting a broad spectrum ICP-MS analysis of the final 165Er solution and an attempt to correlate the resulting trace metal impurities with the results of the DOTA/DTPA AMA titration and [165Er]PSMA-617 radiolabeling experiments. While we do agree that this would be useful information, we did not collect this data for the experiments for a variety of reasons. First, the amount of purified 165Er solution is very small (~400 µL) and entirely consumed through the important analyses described in the manuscript (MP-AES for holmium, DOTA/DTPA AMA titration, and [165Er]PSMA-617 radiolabeling experiments). MP-AES was only performed for holmium and not a broad elemental panel because of limitations related to the total required solution volume for analysis and the detection limits of the instrument. We do not have access to inductively coupled plasma mass spectrometry (ICP-MS) in our laboratory and this would require radioactively decaying solutions and sending them away for analysis, which is logistically challenging and expensive. Additionally, we feel that the most pertinent information related to the quality of the radionuclide preparation is simply whether it is possible to perform high yielding radiopharmaceutical labeling experiments at molar activities that are suitable for targeted radionuclide therapy experiments. The results of the AMA titration (Table 4) and PSMA-617 radiolabeling experiments (Table 5) show this much better than ICP-MS results would, and we believe are sufficient to demonstrate the radiochemical quality of the purified 165
- The reviewer poses a good question. This phenomenon perplexed us for a while, but we eventually determined the radioactivity trace tailing was due to mixing effects resulting from the transition from capillary (0.17 mm inner diameter) tubing leading up to the UV detector to wider bore tubing (0.8 mm inner diameter) occurring before the radiation detector. We are confident in this as we have recently moved the radiation detector to be inline before the UV detector in the HPLC system, eliminating the radioactivity trace tailing. We have added the following explanation to the Supplementary Figure S7 (formerly S6) description, “Mixing effects resulting from the transition from capillary (0.17 mm inner diameter) to wider bore tubing (0.8 mm inner diameter) at the UV detector caused the observed tailing in the downstream radiation detector signal.”
- We agree with the reviewer that automation is an important part of the production of ‘new radionuclides’ for nuclear medicine research applications. Column chromatography is an inherently automatable process and work has begun in our laboratory to automate the 165Er isolation procedure. We have added the following sentence to the Conclusion to highlight this points to the reader, “The separation utilizes column chromatography with commercially available resins and is well suited for automation.”

Round 2
Reviewer 2 Report
The authors answer all my questions and I want to thank them.